# Adapting for the COVID-19 pandemic in Ecuador, a characterization of hospital strategies and patients

Daniel Garzon-Chavez[1]*, Daniel Romero-Alvarez[2,3‡], Marco Bonifaz[4◎], Juan Gaviria[4◎], Daniel Mero[4◎], Narcisa Gunsha[4◎], Asiris Perez[4◎], María Garcia[4◎], Hugo Espejo[4◎], Franklin Espinosa[4◎], Edison Ligña[4◎], Mauricio Espinel[5◎], Emmanuelle Quentin[6‡], Enrique Teran[1‡], Francisco Mora[4◎], Jorge Reyes[4,7]*

1 Colegio de Ciencias de la Salud, Universidad San Francisco de Quito, Quito, Ecuador, 2 Biodiversity Institute and Department of Ecology & Evolutionary Biology, University of Kansas, Lawrence, Kansas, United States of America, 3 OneHealth Research Group, Facultad de Ciencias de la Salud, Universidad de las Américas, Quito, Ecuador, 4 Hospital del Instituto Ecuatoriano de Seguridad Social (IESS) Quito-Sur, Quito, Ecuador, 5 Direccion de Salud Individual y Familiar, Instituto Ecuatoriano de Seguridad Social (IESS), Quito, Ecuador, 6 Centro de Investigación en Salud Pública y Epidemiología Clínica (CISPEC), Universidad Tecnológica Equinoccial, Quito, Ecuador, 7 Facultad de Ciencias Químicas, Universidad Central del Ecuador, Quito, Ecuador

◎ These authors contributed equally to this work.
‡ These authors also contributed equally to this work.
* jorgereyes83@gmail.com (JR); dgarzonc@usfq.edu.ec (DGC)

**Data Availability Statement:** All relevant data are within the paper and its Supporting information files.

## Abstract

The World Health Organization (WHO) declared coronavirus disease-2019 (COVID-19) a global pandemic on 11 March 2020. In Ecuador, the first case of COVID-19 was recorded on 29 February 2020. Despite efforts to control its spread, SARS-CoV-2 overran the Ecuadorian public health system, which became one of the most affected in Latin America on 24 April 2020. The Hospital General del Sur de Quito (HGSQ) had to transition from a general to a specific COVID-19 health center in a short period of time to fulfill the health demand from patients with respiratory afflictions. Here, we summarized the implementations applied in the HGSQ to become a COVID-19 exclusive hospital, including the rearrangement of hospital rooms and a triage strategy based on a severity score calculated through an artificial intelligence (AI)-assisted chest computed tomography (CT). Moreover, we present clinical, epidemiological, and laboratory data from 75 laboratory tested COVID-19 patients, which represent the first outbreak of Quito city. The majority of patients were male with a median age of 50 years. We found differences in laboratory parameters between intensive care unit (ICU) and non-ICU cases considering C-reactive protein, lactate dehydrogenase, and lymphocytes. Sensitivity and specificity of the AI-assisted chest CT were 21.4% and 66.7%, respectively, when considering a score >70%; regardless, this system became a cornerstone of hospital triage due to the lack of RT-PCR testing and timely results. If health workers act as vectors of SARS-CoV-2 at their domiciles, they can seed outbreaks that might put 1,879,047 people at risk of infection within 15 km around the hospital. Despite our limited sample size, the information presented can be

**Funding:** Daniel Romero-Alvarez DRA was supported by a grant from the National Science Foundation (DMS 2028297).

**Competing interests:** The authors have declared that no competing interests exist.

used as a local example that might aid future responses in low and middle-income countries facing respiratory transmitted epidemics.

## Introduction

Originally announced as a pneumonia of an unknown etiology in China [1], the coronavirus disease (COVID-19) had amassed around four million cases worldwide by mid-May 2020 [2, 3]. Ecuadorian implementations to control COVID-19 are relevant as a case study considering its fragmented public health system [4, 5] and the heterogeneous evolution of the pandemic in its different administrative units (i.e., provinces). For instance, two of its main cities, Quito and Guayaquil, applied initial recommended control measures at different times: Guayaquil banned mass gatherings and implemented strict isolation around two weeks later than Quito [6, 7].

The first case of COVID-19 in Ecuador was reported on 29 February 2020 [8]. By 7 April 2020, Ecuador centralized real-time reverse-transcriptase polymerase chain reaction (RT-PCR) testing for SARS-Cov-2 diagnosis in the National Institute of Public Health (INSPI, Spanish). As a consequence, reports were dependent on availability of resources and infrastructure, highly biasing the official case counts [9], misreporting cases, and retaining data due to the lack of processing capacities (S1 Fig; [10]). An initiative to decentralized testing led by university laboratories suggested in 12 March 2020 had the potential to increase RT-PCR diagnosis availability [11], although at 15 April 2020 it was still on plans of implementation [12].

Due to the limitations on testing capacity, on-site approaches for patient triage have been suggested and actively explored using chest computed tomography (CT) and even pure clinical approaches [13, 14]. Chest CT is performed on the majority of hospitalized patients with COVID-19 with main findings including the presence of ground glass opacities (GGO) [15–17]. At least 20% of non-severe COVID-19 infections have shown lack of changes in chest CT scans, while only 3% of severe patients presented normal CTs [15]. Thus, the role of CT in severity screening and diagnosis has been evaluated thoroughly in different parts of the world and recommended as an essential part of COVID-19 diagnosis in different guidelines [18–21].

The Hospital General del Sur de Quito (HGSQ, in Spanish) was inaugurated on 5 December 2017 as a center with 450 beds, providing secondary healthcare with capabilities to solve third level health related complexities. The HGSQ provided medical care to an average of 20,000 monthly patients and performed 1,104 annual surgeries. On 14 February 2020, the hospital was designated to become a COVID-19 specific treatment center. Because this nomination implied that suspected and diagnosed cases of COVID-19 were to be attended exclusively by HGSQ personnel, measures assuring a safe environment for patients and health workers were developed and implemented in a constrained schedule.

### Hospital transition to a COVID-19 specific health center

**Triage strategies based on Artificial Intelligence (AI)-assisted chest computer tomography (CT).** A pandemic event represents a unique challenge for a hospital response, which should mainly focus on preserving the biosafety of patients and health workers, avoid nosocomial infections, and managing typical diseases and chronic patients [22]. As such, the HGSQ took the advantage of an innovative triage approach based on non-contrast chest CT scans to stratify COVID-19 suspected patients according to an artificial intelligence (AI) scoring

system. The software was developed as part of the Huawei Cloud AI services [23] and was implemented for the first time in Latin America for the HGSQ Radiology Department [24].

By calculating internal metrics comparing the predicted lesions from a trained AI (see Methods), with the actual lesions from the CT scan [25], the AI-aided assisted CT screening provided a score that categorized patients in three classes: non-severe (score of 0–30%), moderately severe (30–70%), and severe (>70%), considering the likelihood of being COVID-19 positive in relation with the severity of radiological findings on chest CT [23, 26, 27]. A medical radiologist examined and confirmed these severity scores. Depending on this categorization, patients were distributed in different 'score rooms' across three hospital towers, with a specific flow to prevent the spread of the virus (Fig 1). Similar AI-based approaches for imaging recognition on chest CTs have been deployed in China with contrasting results for COVID-19 diagnosis, the majority of them still requiring further evaluation [19, 28, 29].

## Hospital distribution for COVID-19 patient attendance

Redistribution of hospital areas for COVID-19 attendance occurred in two phases and was guided mainly by the severity scores provided by the AI assisted chest CT scores, due to the difficulties to obtain timely of RT-PCR testing. The Infection Control Unit designated 'score rooms' for different COVID-19 diagnostic categories (Fig 1). Cases with a higher probability of COVID-19 (score >70%) were hosted at Clinic Area 1, patients with moderate probability (score 30–70%) were concentrated at Surgery Area 1. Lastly, patients with a score less than 30%, and therefore a lower likelihood of COVID-19 positivity, were located in Surgery Area 2 and 3 in common rooms separated by gender (Fig 1A). Hospital personnel prevented the CT machine to act as a fomite source of SARS-CoV-2 transmission using disinfection based on conventional cleaning followed by pulsed-xenon ultraviolet room sterilization [30, 31]. A further reorganization was put in place once confirmatory RT-PCR tests became available (Fig 1B), however AI-assisted chest CT scans remained the main approach driving triage considering the lag of laboratory-based testing results, and their application only on patients with higher suspicion of infection. At that point, two elevators were available for patient transportation (Fig 1B). Molecularly confirmed SARS-CoV-2 patients were located exclusively in Clinic Area 1. Surgery Area 1 became exclusively for patients with a score above 70%, surgery area 2 became exclusively for patients with a 30–70% score, as it was Clinic Area 2. Therefore, Clinic and Surgery areas 1 and 2 became SARS-CoV-2 exclusive wards for laboratory confirmed diagnosis and CT-scores >70%. Moreover, other areas of the hospital were accommodated to host patients with a moderate AI-CT score COVID-19 likelihood of infection, including the Gynecology and Obstetrics area, for patients with a score between 30–70% and the Surgery Area 3, divided in two sections: one for patients with a score less than 30% and the other for negative RT-PCR patients in the path of discharge (Fig 1B).

## 'Virus in movement' protocol

To complement the triage approach, from 10 to 28 March 2020, the Unit of Infection Control, in coordination with multiple departments at the HGSQ, developed the 'Virus in Movement (VM)' protocol for COVID-19 crisis management. Every time that a patient suspected of COVID-19 needed transit to any area of the hospital, the VM alert was activated and triggered three coordinated steps: (a) evacuation of all health personnel and patients from the movement areas (e.g., corridors, halls, etc), (b) blocking doors to prevent people transit, and (c) designation of exclusive elevators (Fig 1A and 1B). After movement, areas occupied by suspected/confirmed patients were cleaned with pulsed-xenon ultraviolet room disinfection according to Jinadatha et al. (2015) and Kovach et al. (2017) [30, 31]. The average duration of the

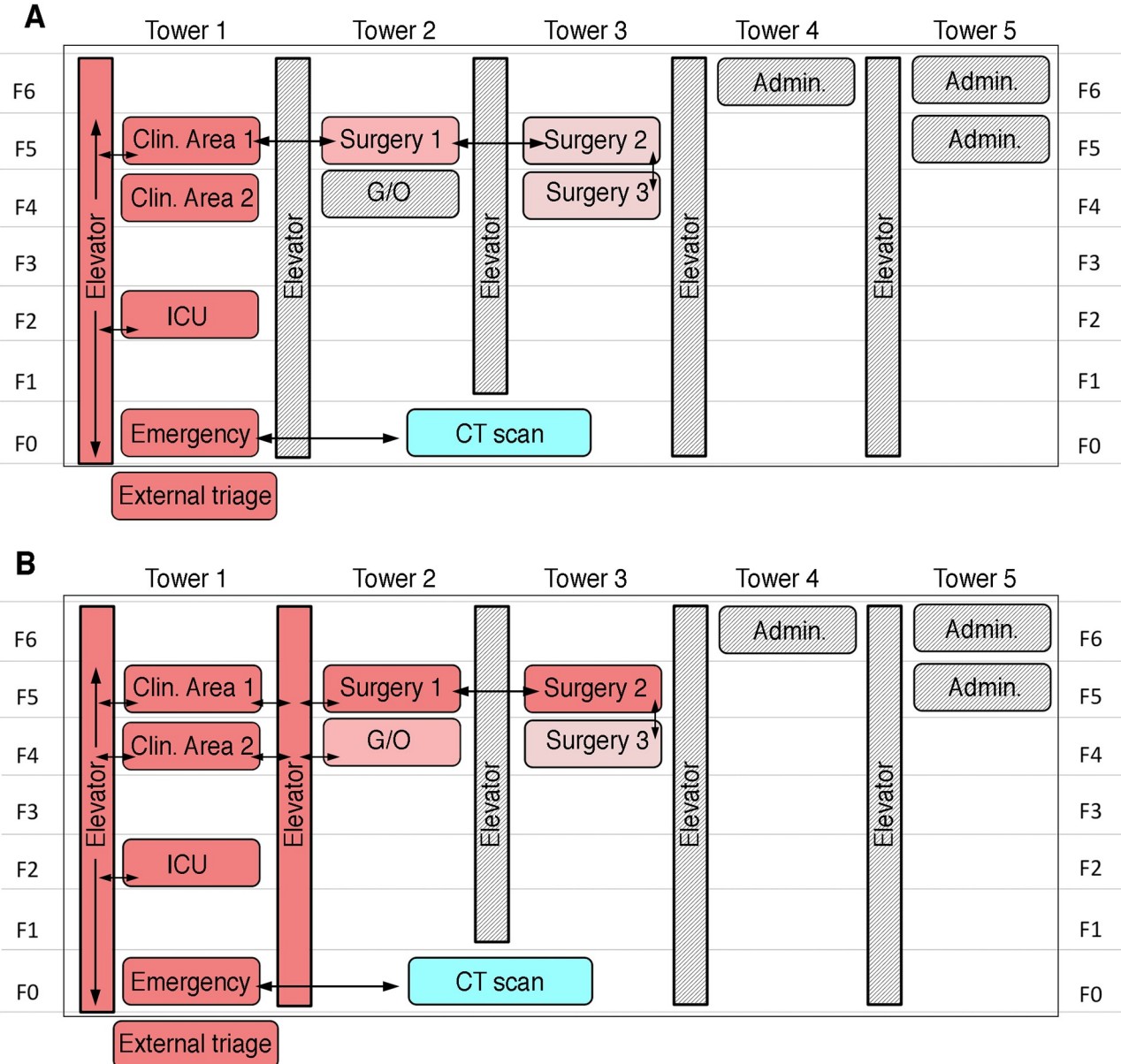

**Fig 1. Schematic representation of the Hospital General del Sur de Quito (HGSQ) highlighting specific COVID-19 designated areas.** The initial distribution of areas dedicated to COVID-19 patients (i.e., score rooms in red; A) was complemented with newer areas to categorize confirmed, suspected, and negative COVID-19 patients based on laboratory diagnosis and a computer tomography (CT) assisted severity score (B; see text). Different red shading represents areas of high and low COVID-19 transmission risk. Arrows (black) summarize the available patient flow around the hospital with only one elevator available during the first time period (A) and two during the second (B). Areas in grey shading were disabled in order to control movement of personnel. The different floors of the hospital are labeled as F0-F6 in both panels. ICU = intensive care unit; G/O = gynecology and obstetrics; Admin. = administrative offices. CT scan = computer tomography scan.

implementation of the full VM protocol was about 1 hour. During the next 18 days of transition, a total of 75 inpatients were transferred to different portions of the hospital a total of 432 times. No health care workers or other personnel transiting within the hospital were infected with SARS-CoV-2 during this period.

In the remainder of the manuscript, we quantify the ability of the AI-assisted chest CT for patient screening and describe the socio-demographic and clinical characteristics of the first

75 patients tested and treated for SARS-CoV-2 at the HGSQ, which represent the first outbreak of Quito city. Moreover, due to the high risk of asymptomatic carriers and superspreader events associated with coronavirus infections [17, 32, 33], we plotted the geographic distribution of 126 health care workers attending COVID-19 cases at the HGSQ, plus 54 laboratory tested COVID-19 cases, to suggest how epidemic surveillance might be organized via geographical information systems (GIS) in Quito, Ecuador.

# Methods

## Ethic statement

This study was reviewed and approved by the Institutional Review Board at Universidad San Francisco de Quito (2020-023M). Information from patients was anonymized before analysis. Patients offered their oral consent for gathering demographic data. Health-care personnel followed intra-hospital guidelines to fill information forms considering their home addresses according to HGSQ policies.

## Artificial intelligent (AI)-assisted chest computed tomography (CT) screening for patients' triage

The Huawei Cloud AI-assisted CT diagnosis software can be described as a deep-learning neuronal network approach for automated medical image segmentation for identification of abnormalities on chest CTs [34, 35], it was released on 17 March 2020 and uses MindSpore as its AI deep-learning algorithm framework, which was developed entirely by Huawei [36, 37]. For calibration purposes, the AI has been trained with ~4,000 chest CT images from confirmed positive COVID-19 cases from China [23]. Scanned CT images were uploaded to the HGSQ picture archiving and communication system (PAC) and then examined with the Huawei AI to detect the presence of GGOs and lung consolidations [23, 25].

Information regarding the Huawei AI-assisted CT screening COVID-19 score system for patient categorization was lacking, with absence of specific technical details about image categorization; thus, we had to implement the system as a black box. We calculated sensitivity and specificity indexes for the Huawei AI-assisted CT screening tool to correctly identify cases as highly likely to be COVID-19 positives (i.e., score >70%) as confirmed by molecular diagnosis via RT-PCR.

## Clinical and epidemiological characteristics of COVID-19 patients

Epidemiological (e.g., hospitalization time, risk factors, source of infection), clinical (e.g., symptoms and signs), laboratory data and drug treatment schemes, were recovered from medical records of hospitalized patients with respiratory symptoms above 18 years old with a RT-PCR test for SARS-CoV-2. We collected data from patients admitted in the HGSQ between 10 to 28 March 2020.

## Spatial distribution of households of patients and health personnel

Health personnel attending inpatients with presumptive or confirmed diagnosis of COVID-19 at the HGSQ completed online forms disclosing attending time, home address, and use of personal protective equipment (PPE) on a daily basis. This was required due to the risk of health workers spreading COVID-19 to other hospitalized patients or the community as asymptomatic SARS-CoV-2 carriers, and the lack of reliable testing for antibody detection [32, 33, 38]. Health workers with symptoms related to COVID-19 were treated by the Department of Occupational Medicine and immediately notified to the Infection Control Unit, to suspend their

activities during 14 days after symptom resolution and RT-PCR negative tests as suggested by different guidelines [20, 21, 39].

We used the information on these forms to identify potential clusters of COVID-19 surveillance outside hospital settings by georeferencing addresses of health workers using Google Maps (https://www.google.com/maps/), calculating the distance to the hospital (i.e., HGSQ), and estimating the amount of people at risk of infection using the 2010–2020 population projections from the official Ecuadorian census (https://www.ecuadorencifras.gob.ec/proyecciones-poblacionales/). We built three buffers of 0–5, 5–10, and 10–15 km distances centered at the hospital and calculated the number of people at risk within each buffer boundary. Distance calculation and population at risk was calculated using TerrSet (version 18.39; https://clarklabs.org/terrset/). Coordinates and results of this analysis were plotted in maps using QGIS (3.4 Madeira; https://qgis.org/es/site/forusers/download).

## Results

### Efficacy of the AI-assisted chest CTs for COVID-19 triage

We obtained chest CTs for 75 patients with laboratory confirmed SARS-CoV-2 diagnosis (Table 1). Images showed that the distribution of the GGOs in the lungs were most peripheral (30/61, 49.18%) than central, the latter detected in 21 of the 61 laboratory-confirmed positive cases (34.43%). Bilateral lesions were predominant. Five laboratory-confirmed infected patients showed an absence of GGOs patterns in the lungs (8.2%). Seven SARS-CoV-2 negative tested cases, showed peripheral GGOs lesions (7/14, 50%) while four showed central GGOs (4/14, 28.57%).

We obtained severity scores for 37 laboratory-tested patients (49.3%). Sensitivity corresponded to 21.4% and specificity to 66.7% when considering the likelihood to classify a patient as COVID-19 positive with a score over 70% (S1 Table). Thus, 7/28 positive and 3/9 negative laboratory-tested cases (n = 10) were allocated in 70% score rooms; 10/20 positive and 1/9 negative laboratory-tested cases (n = 11), in score rooms for the 30–70% category; and 11/28 positive and 5/9 negative laboratory-tested cases (n = 16) were allocated in rooms for scores less 30%.

### Epidemiological characteristics and description of clinical cases

At the moment of data collection, we have attended 2,590 patients with respiratory symptoms, 93 with potential SARS-CoV-2 infection (i.e., 3.5% prevalence) and 18 deaths. We present clinical, epidemiological, and laboratory data, together with treatment schemes for 61 patients with confirmed SARS-CoV-2 infection and 14 patients with a negative test but COVID-19 suspected (n = 75; Tables 1–4). The overall age of patients is 50 years old with a male majority (male/female ratio: 47/28 = 1.67; Table 1).

From the patients with a positive laboratory test, 42.6% (n = 26) reported having traveled to Guayaquil, the city with more cases of COVID-19 in Ecuador during the studied period [9] (Table 1). Moreover, 16.4% (n = 10) laboratory confirmed positive cases reported having a history of close contact with known COVID-19 patients (Table 1). In general, fever, cough, and odynophagia were the most prevalent symptoms, while anosmia was the least common (Table 1). At least ten negative patients also had fever (71.4%) and 13 presented cough (92.9%). One of the negative cases also referred anosmia (7.1%). The average number of days from onset of respiratory symptomatology until hospital attention was eight days (ranging from zero to 20 days).

All cases admitted to the intensive care unit (ICU) presented values of C-reactive protein (CRP) above 10 mg/L and LDH above 250 UI/L. Median values from non-ICU patients also

**Table 1. General characteristics, risk factors, and symptoms/signs of the first laboratory tested COVID-19 cases (n = 75) attended in the Hospital General del Sur de Quito (HGSQ), Quito, Ecuador—March 2020.**

| Patient characteristics | Positive (%) | Negative (%) | Totals (%) |
|---|---|---|---|
| RT-PCR confirmed | 61 (81.3) | 14 (18.7) | 75 (100)* |
| Median age (range) | 50 (25–76) | 49 (27–77) | 50 (25–77) |
| Male | 39 (63.9) | 8 (57.1) | 47 (62.7) |
| Female | 22 (36.1) | 6 (42.9) | 28 (37.3) |
| ICU | 14 (23) | 4 (28.6) | 18 (24) |
| • ICU male | 11 (18.1) | 2 (14.3) | 13 (17.3) |
| • ICU female | 3 (4.9) | 2 (14.3) | 5 (6.7) |
| Non-ICU | 47 (77) | 10 (71.4) | 57 (76) |
| • Non-ICU male | 28 (45.9) | 6 (42.9) | 34 (45.3) |
| • Non-ICU female | 19 (31.1) | 4 (28.5) | 23 (30.7) |
| **Risk factors** | | | |
| Contact with European tourist | 3 (4.9) | 1 (7.1) | 4 (5.3) |
| Contact with COVID-19 case | 10 (16.4) | 4 (28.6) | 14 (18.6) |
| Unknown contact | 14 (23) | 6 (42.9) | 20 (26.6) |
| Travel to Guayaquil | 26 (42.6) | 3 (21.4) | 29 (38.6) |
| Travel to other Ecuadorian cities | 5 (8.2) | 0 (0) | 5 (6.6) |
| Travel to European countries | 2 (3.3) | 0 (0) | 2 (2.6) |
| Health worker | 1 (1.6) | 0 (0) | 1 (1.3) |
| **Symptoms/signs** | | | |
| Fever | 59 (96.7) | 10 (71.4) | 69 (92) |
| Cough | 48 (78.7) | 13 (92.9) | 61 (81.3) |
| Odynophagia | 36 (54) | 5 (35.7) | 41 (54.7) |
| Headache | 13 (21.3) | 2 (14.3) | 15 (20) |
| Diarrhea | 12 (19.7) | 1 (7.1) | 13 (17.3) |
| Vomiting | 5 (8.2) | 0 (0) | 5 (6.7) |
| Anosmia | 4 (6.6) | 1 (7.1) | 5 (6.7) |

Cases categorized as positive/negative by laboratory testing (i.e., RT-PCR) from 10 to 28 March 2020.

*Percentages from the first row are calculated in relation to the total cases (n = 75). Percentages of the following rows are calculated with positives, negatives, and totals from the first row, respectively.

presented higher values for LDH and borderline values of CRP (Table 2). From laboratory positive SARS-CoV-2 cases, those admitted at the ICU presented elevated values of D-dimer than those attended outside the ICU; in both classes the median was higher than normal. The median value for levels of transaminases AST and ALT was elevated only in ICU patients (Table 2).

Procalcitonin was higher in patients admitted to the ICU with lesser values on patients attended outside this unit. Creatinine was within normal range for both ICU and non-ICU cases (Table 2). One case at ICU presented *Candida* spp. in trachea, other patients in the ICU with normal values of procalcitonin presented *Klebsiella pneumoniae* and *Pseudomonas aeruginosa*-antibiotic-sensitive in blood, and *Candida* spp. in urine. Leucocytes and platelet counts for both case categories were within normal values, but lymphocytes were lower for ICU patients (Table 3).

Only two laboratory confirmed negative patients received treatments without chloroquine (Table 4). Most schemes were associated with beta-lactam antibiotics such as ceftriaxone, meropenem, or piperacillin/tazobactam (Table 4). A total of 45/75 patients were discharged. The

**Table 2. Blood chemical values for laboratory-confirmed COVID-19 cases (n = 61) by Intensive Care Unit (ICU) admission status in the Hospital General del Sur de Quito (HGSQ), Quito, Ecuador—March 2020.**

| ICU/Non-ICU | D-Dimer (0–0.5 ng (EUF)/mL) | CPK (<25 UI/L) | LDH (125–243 UI/L) | CRP (<3 mg/L) | PCT (<0.05 ng/mL) | CR (<1.2 mg/dL) | ALT (9–50 UI/L) | AST (15–40 UI/L) |
|---|---|---|---|---|---|---|---|---|
| ICU (n = 14) | 333 (0.09–7567) | 205.5 (35–822) | 481.5 (130–917) | 202 (88.7–395) | 0.43 (0.06–13.9) | 0.88 (0.5–2.85) | 62 (17–95) | 59 (21–117) |
| Non-ICU (n = 47) | 155 (0.11–946 | 68 (16–898) | 272.5 (87–1143) | 48 (2.1–211) | 0.05 (0.01–0.59) | 0.87 (0.53–2.37) | 43.5 (10–240) | 34 (12–106) |

Data collected from 10 to 28 March 2020. Normal ranges of each laboratory parameter are shown in parenthesis in the headers of each column.

Values of each cell represent medians. Ranges are depicted in parenthesis.

CPK = Creatine phosphokinase, LDH = Lactate dehydrogenase, CRP = C-reactive protein, PCT = Procalcitotin, CR = Creatinine; ALT = Alanine transaminase, AST = Aspartate transaminase.

average hospital stay was of 10 days (range = 8–20 days); these patients received the scheme based on azithromycin, ceftriaxone, and chloroquine for 7 days (Table 4).

## Health personnel and COVID-19 cases domiciliary georeferencing

The first COVID-19 case had contact with at least 51 health workers. Our first ten patients had contact with a median of 15 health workers (range = 9–51). After the transition to a COVID-19 exclusive hospital, we reduced contact to a mean of five health personnel (range = 3–7) as those taking care of COVID-19 patients. By georeferencing domiciles of health workers, we identified that 43 live between 0–5 km from the hospital in an urban area with 783,291 inhabitants. Similarly, 39 health workers lived around 5–10 km from the hospital in an area with 647,655 inhabitants. Finally, 22 health workers lived between 10–15 km around the hospital in an area with 448,101 inhabitants, for a total of 1,879,047 people in risk within 15 km around the hospital. The majority of health workers came from the south and northernmost metropolitan area. By this analysis, we also identified that some health workers (n = 4) traveled up to 85 km to get to the HGSQ, crossing other provinces including Imbabura to the north and Cotopaxi to the south (Fig 2).

Domicile georeferencing was completed for 54 out of 75 (72%) patients with COVID-19 positive diagnosis. At the beginning of the epidemic, the majority of patients came from the south of Quito city. The geographic distribution of cases changed once the hospital became exclusive for COVID-19, with cases broadening to encompass the whole Quito metropolitan area and its surroundings including one case arriving from the southern province of Cotopaxi (Fig 2).

## Discussion

Ecuador faced the current COVID-19 pandemic following WHO recommendations [8], however, despite being supported by the central government, main cities managed their health

**Table 3. Leucocyte, lymphocyte, and platelet counts for laboratory confirmed (n = 61) COVID-19 cases by Intensive Care Unit (ICU) admission status in the Hospital General del Sur de Quito (HGSQ), Quito, Ecuador—March 2020.**

| ICU/Non-ICU | Leucocytes (3,500–9,500/ml) | Lymphocytes (1,100–3,200/ml) | Lymphocytes (20–40%) | Platelets count (125,000–350,000/ml) |
|---|---|---|---|---|
| ICU (n = 14) | 8,900 (4,800–15,000) | 1,000 (500–1,800) | 11 (4.2–21) | 183,000 (145,000–277,000) |
| Non-ICU (n = 47) | 5,700 (1,900–13,300) | 1,440 (500–2,490) | 22.5 (8.9–61.7) | 210,000 (114,000–452,000) |

Data collected from 10 to 28 March 2020. Normal ranges of each laboratory parameter are shown in the headers of each column.

Values of each cell represent medians. Ranges are depicted in parenthesis.

**Table 4. Treatment schemes applied to laboratory confirmed COVID-19 cases (n = 75) by Intensive Care Unit (ICU) admission status in the Hospital General del Sur de Quito (HGSQ), Quito, Ecuador—March 2020.**

| Therapy | Laboratory confirmed positive (n = 61) | Laboratory confirmed negative (n = 14) | Totals (positive + negative, n = 75) | Hospitalized (n = 28) | Hospital discharged (n = 47) |
|---|---|---|---|---|---|
| **ICU** | | | | | |
| AZT + CEF + CLOR | 1 | - | 1 | 1 | - |
| AZT + CEF + CLOR + LOP/ RITO + OSEL | 3 | - | 3 | 3 | - |
| AZ + CLOR + LOP/RITO + OSEL + PIP/TAZ** | 8 | 4 | 12 | 7** | 2 |
| CLA + CLOR + LOP/RITO + PIP/TAZ | 2 | - | 2 | 2 | - |
| Totals | 14 (22.9%) | 4 (28.6%) | 18 (24%) | 13+3** = 16 | 2 |
| **Non-ICU** | | | | | |
| AZT + AMOX/CLA + CLOR | 1 | - | 1 | - | 1 |
| AZT + CEF + CLOR | 38 | 8 | 46 | 10 | 36 |
| AZT + CLOR + LEV | 2 | - | 2 | - | 2 |
| AZT + CEF + CLOR + LOP/ RITO | 3 | - | 3 | - | 3 |
| AZT + CEF + CLOR + LOP/ RITO + OSEL | 1 | - | 1 | 1 | - |
| AZT + CLOR + LOP/RITO + MER + OSEL | 1 | - | 1 | 1 | - |
| AZT + CLOR + LOP/RITO + OSEL + PIP/TAZ | 1 | - | 1 | - | 1 |
| CEF + CLA | - | 1 | 1 | - | 1 |
| LEVO | - | 1 | 1 | - | 1 |
| Totals | 47 (%) | 10 (%) | 57 (%) | 12 | 45 |

Outputs considered up to the end of data collection for intensive care unit (ICU; n = 14) or non-ICU (n = 47) cases that remained in the hospital or were discharged.

Percentages are calculated in relation to the total number of laboratory confirmed cases (n = 75).

AMOX/CLA = Amoxicilin/Clavulanic Acid, AZT = Azithromicyn, CEF = Ceftriaxone, CLAR = Clarithromycin, CLOR = Chloroquine, LEVO = Levofloxacin, LOP/ RIT = Lopinavir/Ritonavir, MER = Meropenem OSEL = Oseltamivir, PIP/TAZ = Piperaciline/Tazobactam.

**Three patients with this treatment scheme died.

crisis differently. Despite the National Emergency Operations Committee (COE, Spanish) recommended social isolation and stopping all mobility on 16 March 2020 [6], Guayaquil had an underemployed working population of 16.2% [40] with urbanistic features (e.g. slums and lack of water provision) that complicated the implementation of these measures [41]. Quito stopped all face-to-face academic activities earlier than the COE recommendations (12 March 2020; [42, 43]), and followed guidelines of constrained mobility and mass gatherings strictly, buying the HGSQ time to develop the aforementioned approaches, which effectively allowed it to transit to a COVID-19 exclusive attention center. In the hospital, strategies of management, including change in distribution of suspected COVID-19 patients (Fig 1), has allowed us to halt the high incidence of nosocomial infections reported elsewhere [17], which remained in zero until data collection (28 March 2020).

This is the first time that the Huawei Cloud AI-assisted CT screening for COVID-19 is assessed as a categorization tool in Ecuador at the HGSQ, which received the software for free [27]. The same tool has been deployed for hospitals in other countries including China, Malaysia, and the Philippines [44, 45]. For our particular case, the tool was used for initial triage of suspected patients due to the lag between RT-PCR testing and result availability (range 28–120

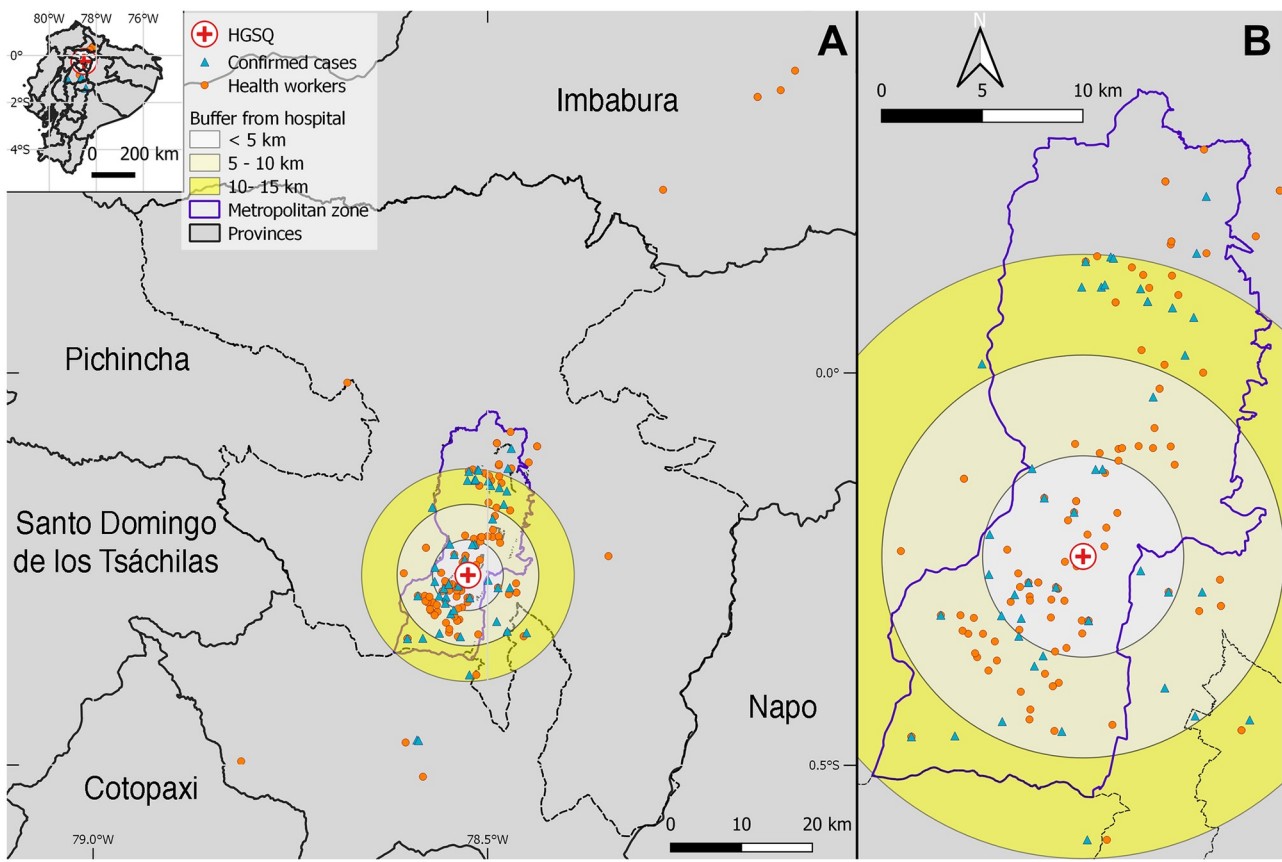

**Fig 2. Distribution of 54 COVID-19 cases and 126 health workers from the Hospital General del Sur de Quito (HGSQ), in Quito, Ecuador.** Buffers represent 0–5, 5–10, and 10–15 km distances around the HGSQ (red cross) and encompass the amount of population living in those areas according to Ecuadorian estimates for 2020. (A) Some health care workers (orange circles) traveled across different provinces to the HGSQ. For the date of data collection, COVID-19 cases (blue triangles) came from different regions of the entire metropolitan zone of Quito municipality (B).

hours), concentrating all efforts to stop a potential COVID-19 nosocomial outbreak. However, even for the most severe cases, with a higher likelihood of COVID-19 infection, we obtained unacceptable test accuracy values for sensitivity and specificity (i.e., 21.4% and 66.7% respectively). This has two immediate consequences, first, from a research perspective, the performance of the AI-assisted CT screening for COVID-19 using Huawei technology is too poor to recommend, something that has also been cautioned by the Philippine College of Radiology [46]. Second, from a pragmatic perspective, the HGSQ continue to rely on this method for patient triage because it relieves a health care bottleneck in the current burgeoning epidemic: patients can be allocated to specific areas depending on the severity score to prevent COVID-19 spread (Fig 1), a crucial endeavor considering the potential correlation between increasing health burden and mortality [47].

In order to fully test the ability of this AI-based screening system, a strict study design should evaluate both laboratory confirmed positive and negative cases [15, 18, 19, 48]. We currently lack information from the latter since we cannot justify using the unique CT facilities of the hospital to expose non-suspected patients to infection. Nevertheless, during the development of this study, zero cases have been associated with exposure of patients or health workers to the CT area, which shows preliminary evidence that the routine cleaning plus pulsed-xenon ultraviolet disinfection approaches, effectively prevent the CT to become a fomite source of

SARS-CoV-2 contagion [30, 31]. It is important to emphasize that the Huawei Cloud AI-assisted CT screening software was donated freely to the HGSQ and we took the advantage to scientifically evaluate it. Currently, there are many other AI-medical image processors that have been deployed, especially in China, and that merit further assessment as well [28].

Patients attended here were part of the first COVID-19 outbreak in Quito. Despite our limited sample of laboratory confirmed positive cases (n = 61), demographic and clinical characteristics of COVID-19 infection were similar to that of previous reports [17, 49, 50], namely, a majority of male individuals with a median age of 50 years presenting fever, cough, and odynophagia (Table 1). Although less frequent, we also found cases describing anosmia, which has been correlated with positive COVID-19 infections [51]. In the present study we did not find cutaneous manifestations of the disease [52]. Blood chemical markers such as CRP, LDH, CPK, etc., were elevated in patients admitted to ICU in comparison with non-ICU patients (Table 2). Values for D-dimer, were above normality for ICU and non-ICU patients but higher for the former than the latter (Table 2); coagulopathies have been incriminated as drivers of mortality for patients with a laboratory confirmed COVID-19 infection [17, 53]. From our variables considered for blood count, only lymphopenia was apparent for ICU cases (Table 3).

In Ecuador, RT-PCR laboratory testing for SARS-CoV-2 was centralized in few hospitals, institutes, and private laboratories, up to 46 days since the beginning of the epidemic [12]. This factor influenced diagnosis delay, epidemic spread, and the urgency to implement out-of-the-box triage approaches such as the one presented here [13, 14, 54]. By the time of this study, testing was limited to 7.46 per 10,000 inhabitants [4], thus, clinical suspicion of patients with respiratory symptoms with elevation of CRP, LDH, and lymphopenia (Tables 2 and 3), can be useful markers for triage in settings unable to rely on molecular or radiological tests [13, 14].

In this study, the majority of laboratory confirmed cases received treatment schemes based on the combination of chloroquine plus azithromycin except for two negative cases, which received clarithromycin plus ceftriaxone or levofloxacin (Table 4). Gautret et al. (2020), showed that the treatment based in hydroxychloroquine plus azithromycin was associated with viral load reduction/disappearance in a small sample size of COVID-19 patients [55]. Two recent clinical trials assessing the safety and efficacy of hydroxychloroquine and chloroquine recommended to avoid COVID-19 treatment schemes with any of these drugs due to the detected increased mortality and lack of benefit [56–58]. Regardless, a more recent study showed a lack of association between treatments including hydroxychloroquine and development of poor clinical outcomes [59]. We were unable to investigate electrocardiogram QT alterations as previously reported [60]. Due to the lack of control groups, our findings should be interpreted as preliminary and by no means as evidence to support any treatment scheme, which is still a topic largely debated with no consensus [21]. Lopinavir/Ritonavir treatment was exclusively used in ICU patients. A case control study published on 18 March 2020, suggested a lack of effect in death reduction [61, 62], however the ICU Department from this hospital decided to continue with the antiviral treatment scheme due to the lack of literature consensus and an apparent clinical improvement still on quantification.

During the progression of the epidemic in Ecuador, officers from the Ministry of Public Health have reported ~1,500 health workers getting infected with SARS-CoV-2 while minimizing the need of full body personal protection [63]. Assuming that infected health workers might act as vectors of SARS-CoV-2 [32], we estimated that at least 1,879,047 people within our designated buffers at the metropolitan area of Quito, might be at risk of infection (Fig 2). Our findings reiterate the need to protect health workers and provide them adequate personal protective equipment to avoid seeding COVID-19 outbreaks [64]. We believe that novel surveillance approaches as the one suggested here should be leveraged and encouraged to complement efforts of epidemiological surveillance to better improve epidemic control.

Our manuscript presents data from the first COVID-19 outbreak in Quito attended in the HGSQ. Despite our limited sample size, we offer a local perspective on how a hospital might be reorganized in a limited amount of time to respond against an emergent epidemic. Further, we show how in spite of its actual limited capacity to discriminate infectious individuals, the AI-assisted chest CT became a key component of the HGSQ transition and triage strategy (i.e., sensitivity = 21.4%). Hospitals in low and middle-income countries might follow a similar approach if there is a lag of evidence-based information with respect to actual response needs. Future analysis should include larger samples sizes and should be shared in a timely matter.

It is important to note that the case of the HGSQ is uncommon in comparison to other Ecuadorian health facilities. Health sector in Ecuador is fractured, encompassing public, social security, military, police, and private health providers [5]; thus, hospital management and protocols are far from standardized [65]. The information published here might aid the implementation of protocols in other regions of Ecuador and also other regions of Latin America, where overrun health systems aiming to control the current epidemic, or potentially future emergent respiratory transmitted diseases, are in need of local perspectives [66].

## Supporting information

**S1 Fig. Official cumulative and daily case counts of COVID-19 in Ecuador from 29 February to 24 April 2020.**
(DOCX)

**S1 Table. Contingency table evaluating Artificial Intelligence (AI)-assisted chest computer tomography (CT) system for COVID-19 triage.**
(DOCX)

**S2 Table. Daily and cumulative number of cases per province in Ecuador from the first case on February 29th to April 24th, 2020.**
(XLSX)

**S3 Table. Clinical, epidemiological, radiological, and treatment data for 75 patients attended at the Hospital General Sur de Quito (HGSQ) between March 13th to March 28th, 2020.**
(XLSX)

**S4 Table. Chemical values and white-blood cell counts for COVID-19 patients attended outside of the intensive care unit (non-ICU) from the Hospital General del Sur de Quito (HGSQ), Quito, Ecuador.**
(XLSX)

**S5 Table. Chemical values and white-blood cell counts for COVID-19 patients attended at the Intensive Care Unit (ICU) from the Hospital General del Sur de Quito (HGSQ), Quito, Ecuador.**
(XLSX)

## Acknowledgments

The authors acknowledge Jose Hector Cadena who helped us with English revisions.

## Author Contributions

**Conceptualization:** Daniel Garzon-Chavez, Jorge Reyes.

**Data curation:** Daniel Garzon-Chavez, Marco Bonifaz, Juan Gaviria, Daniel Mero, Narcisa Gunsha, Asiris Perez, María Garcia, Hugo Espejo, Franklin Espinosa, Edison Ligña, Mauricio Espinel, Francisco Mora.

**Methodology:** Emmanuelle Quentin.

**Writing – original draft:** Daniel Garzon-Chavez, Jorge Reyes.

**Writing – review & editing:** Daniel Romero-Alvarez, Enrique Teran.

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
