## [Decision Letter · Decision Letter 0]

5 Feb 2021

PONE-D-20-15485

Adapting in middle of COVID-19 pandemic in Ecuador, a Hospital strategies and patients characterization

PLOS ONE

Dear Dr. Garzon,

Thank you for submitting your manuscript to PLOS ONE. After careful consideration, we feel that it has merit but does not fully meet PLOS ONE’s publication criteria as it currently stands. Therefore, we invite you to submit a revised version of the manuscript that addresses the points raised during the review process.

The manuscript suffers for a lot of bias that should be addressed to have chance to be published. In particular, English Language needs a strong revision by a native english speaker. Furthermore, it needs a major reorganization: in particular, the sections on how the hospital was transformed into a COVID-19 reference hospital, as well as protocol changes in patient management should be moved to the introduction section from the methods and the results.

We look forward to receiving your revised manuscript.

Kind regards,

Adriana Calderaro

Academic Editor

PLOS ONE

Journal Requirements:

4. We note that Figure 3 in your submission contains map images which may be copyrighted. All PLOS content is published under the Creative Commons Attribution License (CC BY 4.0), which means that the manuscript, images, and Supporting Information files will be freely available online, and any third party is permitted to access, download, copy, distribute, and use these materials in any way, even commercially, with proper attribution. For these reasons, we cannot publish previously copyrighted maps or satellite images created using proprietary data, such as Google software (Google Maps, Street View, and Earth). For more information, see our copyright guidelines: http://journals.plos.org/plosone/s/licenses-and-copyright.

(1) You may seek permission from the original copyright holder of Figure 3 to publish the content specifically under the CC BY 4.0 license. 

Reviewers' comments:

Reviewer's Responses to Questions

**Comments to the Author**

1. Is the manuscript technically sound, and do the data support the conclusions?

Reviewer #1: Partly

2. Has the statistical analysis been performed appropriately and rigorously? 

Reviewer #1: N/A

3. Have the authors made all data underlying the findings in their manuscript fully available?

Reviewer #1: Yes

4. Is the manuscript presented in an intelligible fashion and written in standard English?

Reviewer #1: No

5. Review Comments to the Author

Reviewer #1: Overall, I think this manuscript is promising and contains important information for other hospitals in LMIC may find useful in managing COVID-19 patients (specifically the triaging of patients and the software used to identify potential cases). The study is somewhat ambitious and covers many topics and can feel "scattered" at times. However, with some reorganization, I think this would make for a strong descriptive paper. Please see the attached document for specific comments by section.

1. Is the manuscript technically sound, and do the data support the conclusions?

Partly

As this is a descriptive paper without regression modeling, the methods are adequate, but the authors should describe the study samples a bit more clearly, as well as tue quantitative methods for determining the reliability of the AI-assisted CT chest scans. I have included specific comments in the attachment.

*2. Has the statistical analysis been performed appropriately and rigorously?

N/A

*3. Have the authors made all data underlying the findings in their manuscript fully available?

Yes

*4. Is the manuscript presented in an intelligible fashion and written in standard English?

No

I strongly recommend that the authors use professional English language editing services prior to submitting their revision. The errors are too many to list here. There are several instances of uncommon terminology or inconsistent usage (for example, "altered" instead of "elevated" when discussing elevated lab values), or "department" vs "service" when discussing specific departments within the hospital. As an intermediate Spanish speaker, I was able to understand some sentences that appeared to be literal translations from Spanish to English, but those who do not understand Spanish may not be able to glean some of these contexts.

6. PLOS authors have the option to publish the peer review history of their article (what does this mean?). If published, this will include your full peer review and any attached files.

Reviewer #1: No

---

## [Author Response · Author response to Decision Letter 0]

2 Mar 2021

Response to reviewers 1

Title: Adapting for the COVID-19 pandemic in Ecuador, a characterization of hospital strategies and patients

Authors: Daniel Garzon-Chavez, Daniel Romero-Alvarez, Marco Bonifaz, Juan Gaviria, Daniel Mero, Narcisa Gunsha, Asiris Perez, María Garcia, Hugo Espejo, Franklin Espinosa, Edison Ligña, Mauricio Espinel, Emmanuelle Quentin, Enrique Teran, Francisco Mora, Jorge. Reyes

NOTE: Lines mentioned in this response refer to the version of the manuscript with track changes.

Reviewer #1

Comment 1. The description of the creation of the COVID-19 management center should be moved to the introduction section, along with the protocol for triaging patients.

Answer: We thank the reviewer for this recommendation. We have moved the corresponding sections to introduction (see lines starting 177). 

Comment 2. The methods section should be devoted strictly to the quantitative methodology (assessment of the sensitivity and specificity of AI-assisted CT software, description of the patients, and geo-distribution of health care workers), as should the results section.

Answer: We agree with the reviewer, in this version, methods section only describes the three mentioned categories with the entire protocol of patient classification transferred to the introduction (see lines starting 177 and methods section). 

Comment 3. I recommend that the abstract be revised so that it contains all the expected parts of an abstract (brief introduction, purpose/objective, methods, results discussion). The current abstract outlines the purpose of the study but leaves out study results and conclusions. At 178 words, the authors should have room to make use of the full 300 words allotted to include these important findings, such as the sensitivity/specificity of the imaging software to detect true cases, and other salient findings.

Answer: We thank the reviewer for this comment and agree with him/her. The new version has 284 words and includes the most important results obtained in the manuscript (see abstract).

Comment 4. I recommend moving the protocol for hospital transformation and patient management into the introduction section. Sub-section headings will be helpful here.

Answer: We have moved this section as recommended and also used specific subheadings. Thanks! (see lines starting 177). 

Comment 5. Line 58: Please revise. Is this referring to 97,209 cumulative cases or 97,209 cases reported in the three days following the release of 2,195 cases? The figures do not support the 97,209 number mentioned here.

Answer: We apologize with the reviewer for the typo. There was an extra space between 97 and 209 that was missed in the final edition of the manuscript. We were trying to emphasize how, after the release of 2,195 cases, official case counts only reported 97, and then 209, and then 63 cases. Because this information is reporting early states of the epidemic in Ecuador, we have moved this section to the supplementary material. Thanks! (see supplementary material). 

Comment 6. Line 60: These figures are not clear. Data are shown on two scales: Bars for daily case counts and dots for cumulative cases. Neither scale goes up to 22,000

Answer: We agree with the reviewer, none of the scales reach 22,000 cases because, as mentioned in the text, we were depicting the epidemiological curves up to one day before the release of the 11,000 cases. We have amended this problem and now, by including the data for 24 April 2020, supplementary figure 1A shows the correct scales. As mentioned before, we have moved this information to the supplementary material. Thanks! 

Comment 7. The authors should briefly describe the study participants, as there are two groups (patients and health care workers). For patients, please clarify the inclusion criteria; several dates were mentioned in the introduction section, but I was not sure what the dates meant. The text (line 157-158) reads, “We considered data from patients either death or discharged admitted in the HGSQ during February 29th to March 28th.” Does the Feb 29-Mar 28 range refer to the admission date or the death/discharge dates?

Answer: The present study has only one group for the COVID-19 epidemiological assessment: patients above 18 years with respiratory symptomatology, and laboratory testing. We collected 75 cases meeting these criteria between 10 to 28 March. The previous date, 29 February 2020, referred to the first COVID-19 patient detected in Ecuador and was the source of the confusion. We have amended this wording across the manuscript (see lines 423 and 716-717). 

Comment 8. The author should briefly outline how they calculated the reliability of the AI-assisted CT scan software.

Answer: Thanks for this observation. Sadly, the AI-assisted chest CT software has been presented as a ‘black box’ deep learning algorithm with no technical details on the parameters chosen to develop the scoring system. We now have emphasized this detail in the methods section (see lines 412-414). 

Comment 9. Line 149: Unclear what is meant by a “score room,” although further below, in the results, the authors described how patients were separated by score into 3 categories, so I’m assuming that’s what that means. Moving the narrative on patient management and triangulation from the results to the introduction should help clarify things.

Answer: We have followed these recommendations providing more information regarding score rooms and moving all this description to the introduction section. Thanks! (see lines starting 177). 

Comment 10. In the sub-section on spatial distribution (beginning line 158), please also include the frequency that information was collected from the health-care workers. Did they complete the online forms daily, weekly, or just once over the study period (lines 163-164)? 

Answer: Health care workers collected this information daily. We have added this detail in the main text. Thanks! (see line 427).

Comment 11. Line 172: Please list the distances for each buffer zone.

Answer: We have added this information in the figure, the methods section and the legend of the figure. Thanks for the suggestion (see lines 525-526, 975-976, and Fig. 3).

Comment 12. Description of the creation of the COVID-19 management center (lines 194-236) should be moved to the introduction.

Answer: We have moved this section to introduction (see lines starting 177). 

Comment 13. Line 200: Unclear what is meant by “the VM protocol was activated four times.” Does it mean there were 4 patients that needed to be managed in the first 24 hours? Without context/knowing what the average or expected number of activations is (for example—is a higher number good or bad?), this number is somewhat meaningless.

Answer: We agree with the reviewer. We have eliminated this number and now we are plainly describing how it was implemented (see lines 268-272).

Comment 14. Line 207 and throughout: Please be consistent in using the term “service” or “department.”

Answer: We have eliminated the word service and are using department across the text. Thanks!

Comment 15. Line 240 and throughout: Please be consistent in the decimal points chosen when expressing a number. Sometimes 1 decimal point is used and sometimes 2. For percentages, 1 decimal point is enough precision.

Answer: We have reviewed the manuscript and decided to use 1 decimal point as recommended by the reviewer. Thanks!

Comment 16. Lines 253-260: There should be a 2 x 2 table to accompany the information in this paragraph.

Answer: We agree with the reviewer, but also believe that the text is conveying the information accurately. Regardless, we have included the table for sensitivity and specificity calculation as part of the supplementary material. Thanks! 

Comment 17. Line 267: Table shows median, not mean, but text says “mean.” Please confirm which one is the correct one.

Answer: We have corrected to ‘median’ in the main text which is the statistic used for this description (see line 723). 

Comment 18. Line 270: Figure 1 is referred to in the sentence on the % of people who travelled to Guayaquil, but that figure doesn’t convey that information, Table 1 does. In addition, Supplementary file 1 appears to contain raw data, which is not necessary to refer to in the narrative.

Answer: Thanks for noticing this problem. We now refer the reader to Table 1. Thanks! (see line 807). 

Comment 19. Line 280 and throughout: The authors use the term “altered” but I think what they meant was “elevated.”

Answer: We have reviewed and changed the verbose across the manuscript. Thanks! (see lines 830 and 832). 

Comment 20. Line 320: Should be “health personnel.”

Answer: Amended (see line 944)

Comment 21. Line 321: Median may be more appropriate here than mean.

Answer: We agree with the comment. We have changed the statistic as indicated (see line 946). 

Comment 22. Line 324: Not clear if these were all health workers or just the ones who took care of COVID-19 patients.

Answer: Only those in charge of COVID-19 patients. We have added this information (see line 948). 

Comment 23. Overall comment: Ensure that the captions of all tables and figures are descriptive enough such that readers can understand the content of the tables and figures without having to read the text. This usually means describing what the table shows (“distribution of…” “sensitivity and specificity of…” etc.), any other variables that are being crosstabbed, the sample/sub-sample and sample size, and dates (if applicable).

Answer: Thanks for pointing this detail. We have made the legends of tables and figures more descriptive and now the reader will be able to understand them without need of the main text (see figures and tables). 

Comment 24. Please use boldface for table superheaders, headers, and sub-headers (within table) so readers can tell easily where different sections of a table starts/ends.

Answer: We have applied this suggestion for tables in the main text and supplementary material (see tables). 

Comment 25. Removing vertical lines and minimizing the amount of horizontal lines in the table would improve its readability.

Answer: Amended (see tables). 

Comment 26. Table 1. The caption needs to be revised as the content of this table is unclear. The narrative states there were 75 patients with laboratory-confirmed SARS-CoV-2 infection, the column headers say “positive” and “negative,” so are these referring to the positive or negative determination based on the chest CT scan? If so, this needs to be stated explicitly, as well as the number of people (75)

Answer: We agree with the reviewer and for this version we have amended the confusing wording. We referred to laboratory tested for SARS-CoV-2, not positive for infection. As shown in Table 1, the column ‘positive’ and ‘negative’ refers to the status of the patient according to their laboratory RT-PCR result; we have emphasized this detail in the legend as well. Patients from tables 1-4 refer to those either positive/negative based on RT-PCR. The strategy based on AI-assisted chest CT was used only for triage purposes (see table). 

Comment 27. Table 1. The first column needs a header. If these are patient characteristics, that needs to be stated.

Answer: We have added the suggested header. Thanks (see table 1). 

Comment 28. Table 1. There needs to be a superheader for the column stating exactly what “positive” and “negative” refer to (lab test or CT scan scoring result?).

Answer: We have clarified the positive/negative status in the legend. They refer to SARS-CoV-2 laboratory tested cases (see table 1 legend and comment 26). 

Comment 29. Table 1. Please check the percentages in this table again. For the row on RT-PCR confirmation, that’s clear that row percentages are shown. However for male and female (for example), it is not. 63.9 + 57.1 > 100%, while 57.1 + 42.3 does not add up to 100%.

Answer: In the legend we specify that for the RT-PCR confirmed cases, we calculate percentages based on the total of that row, and how for the rest of the table, we used the totals of each positive, negative and totals from the first row to calculate percentages. Thanks for noticing the typo for the percentages of male and females; it was 57.1 + 42.9 = 100%, respectively. We have reviewed all cells and now we have made sure that no errors are present in the table. Thanks! (see table 1). 

Comment 30. Table 2. Please revise the caption to be descriptive of the information contained in the table, including the sample size (e.g. “Blood chemical values comparing XXX to XXX among YYYY patients (n=ZZZ), dates”).

Answer: We have updated the table caption as suggested. Thanks! (see table 2). 

Comment 31. Table 3. It would be helpful to use thousand separators for the numbers (e.g. 2,000 instead of 2000).

Answer: We have amended this detail across the manuscript. Thanks! (see table 3). 

Comment 32. Table 3. Same comment as above to revise the caption to be descriptive of the information contained in the table.

Answer: Amended (see table 3). 

Comment 33. Table 4. Same comment as above to revise the caption to be descriptive of the information contained in the table.

Answer: Amended (see table 4). 

Comment 34. Table 4. See my comment on formatting the tables using boldface and minimize use of lines in the table to make it easier to read. In particular, the sub-sections on ICU and non-ICU patients should be made more prominent.

Answer: All the suggestions have been implemented (see table 4). 

Comment 35. Table 4. The column headers on “remain or transfer to ICU” and “hospital discharged” should have (n)s listed. Please also clarify what “remain” means. Are these patients that remained in their score room? Perhaps the people who remained in their score room should be presented separately from people who were transferred to the ICU (e.g. split the data in this column into two separate columns).

Answer: We apologize with the reviewer for the confusing headers used for the previous version of the manuscript. For the current version we have clarified patient status to ‘Hospitalized’ since we were referring to patients that remained hospitalized at the end of the study period. They are already stratified as ICU or non-ICU in the table. Thanks! (see table 4). 

Comment 36. Table 4. It may be helpful to list treatment regimens that contain the same medications close to one another, but in the order of increasing medicine. For example, if the treatments can contain medicines A, B, C, and D, list A + B first, then list A + B + C or A + B + D. That way, the common medications are obvious, so other clinicians easily see what other medications are in other regimens.

Answer: Thank you very much for this comment. We have reordered drug names by alphabetical order and rearranged the rows to reflect the progressive inclusion of an additional drug. By this way, treatment schemes are now more traceable (see table 4). 

Comment 37. Table 4. -It may also be helpful to list these in order of descending frequency if the goal is to show which treatment regimen was most common for ICU and non-ICU patients. Note that this recommendation may conflict with the recommendation immediately above, so the authors should decide on which presentation would be most helpful to other clinicians.

Answer: See comment 36. 

Comment 38. Figure 1. I recommend giving a letter to each of the 4 panels and referring to the panels specifically, for example “Figure 1A,” so that it is clear which one is being referenced.

Answer: We have added a letter to each panel and legend now refers to these panels. We have moved the table to the supplementary material since the information is outdated (see supplementary material). 

Comment 39. Figure 2. Ensure correct English spelling of room names

Answer: We have redesign Figure 2 to improve color-wise interpretability and fix the errors regarding English name of different hospital sections. Thanks! (see figure 2). 

Comment 40. Figure 3. The colors on this map are hard to read, especially for people with color blindness. Recommend using different shapes instead of different colors. 

Answer: Thanks for pointing this detail. We have updated the figure to use colorblind safe colors (see figure 3). 

Comment 41. Figure 3. The hospital marker needs to be more prominent. 

Answer: We agree with the reviewer, the marker for the hospital is clearer in this version (see figure 3). 

Comment 42. Figure 3. I strongly recommend including the distance for each buffer zone radius somewhere in the figure

Answer: We have added buffer distances in the legend of the figure together with different shades of yellow to denote distances with respect to the hospital. Thanks! (see figure 3). 

Comment 43. Overall, the discussion was well-organized, and synthesized information from the results with the current literature. The authors should expand upon their discussion of geographic risk of populations exposed to health care workers caring for COVID-19 patients (lines 432-438), with an emphasis on the population at risk in the metro area.

Answer: We have added a paragraph emphasizing the number of people at risk within Quito Metropolitan Area due to the potential infected health workers seeding external COVID-19 outbreaks. Thanks! (see lines 1107-1113). 

Comment 44. The authors should include a paragraph describing the limitations of the current study.

Answer: We have noticed our limitations in a paragraph as suggested (see lines 1135-1143). 

Comment 45. The manuscript should conclude with a discussion of the strengths of the current study, as well as what future studies should consider.

Answer: We have included the strengths of our manuscript in a paragraph as suggested (see lines 1135-1143). 

Comment 46. The references appear to be up-to-date. However, the authors need to ensure that the formatting follows Journal guidelines. 

Answer: We have reviewed the references making sure that they follow PLoS One guidelines. (see references). 

Comment 47. References for internet sources of information, including news websites, or any official websites (ministry of health, WHO, PAHO, etc.) include the URL and the date that the resource was accessed.

Answer: We have reviewed the references from Internet sources and now all of them include its corresponding URL and date of access. Thanks! (see references). 

Comment 48. Ensure consistency of spelling throughout (COVID-19 vs COVID19).

Answer: In this version we are using COVID-19 for the entire manuscript. Thanks! 

Comment 49. Recommend professional English language editing for grammar and spelling.

Answer: We have improved the English correcting multiple sections of the manuscript, tables, and figures, and revising the entire text by a native English speaker. We believe that the current version has good English quality. 

Comment 50. For dates, no comma after month (e.g. “December 2019”, not “December, 2019”).

Answer: We have formatted the date to use day month and year without commas (e.g., 28 February 2020) across the manuscript. Thanks!

---

## [Decision Letter · Decision Letter 1]

6 Apr 2021

PONE-D-20-15485R1

Adapting for the COVID-19 pandemic in Ecuador, a characterization of hospital strategies and patients

PLOS ONE

Dear Dr. Garzon,

Thank you for submitting your manuscript to PLOS ONE. After careful consideration, we feel that it has merit but does not fully meet PLOS ONE’s publication criteria as it currently stands. Therefore, we invite you to submit a revised version of the manuscript that addresses the points raised during the review process.

The Authors have addressed to all the comments, however they should amed some grammar errors as suggested by the Reviewer.

We look forward to receiving your revised manuscript.

Kind regards,

Adriana Calderaro

Academic Editor

PLOS ONE

Journal Requirements:

Reviewers' comments:

Reviewer's Responses to Questions

**Comments to the Author**

1. If the authors have adequately addressed your comments raised in a previous round of review and you feel that this manuscript is now acceptable for publication, you may indicate that here to bypass the “Comments to the Author” section, enter your conflict of interest statement in the “Confidential to Editor” section, and submit your "Accept" recommendation.

Reviewer #1: All comments have been addressed

2. Is the manuscript technically sound, and do the data support the conclusions?

Reviewer #1: Yes

3. Has the statistical analysis been performed appropriately and rigorously? 

Reviewer #1: Yes

4. Have the authors made all data underlying the findings in their manuscript fully available?

Reviewer #1: Yes

5. Is the manuscript presented in an intelligible fashion and written in standard English?

Reviewer #1: No

6. Review Comments to the Author

Reviewer #1: I am very pleased with the authors’ responsiveness to my earlier comments. I appreciate the authors’ taking the time to address all of the reviewer's concerns. I believe the flow of the narrative is much clearer after the reorganization. I do not have any major concerns, but I have specific grammar comments, as the PLOS One review system asked reviewers to point out those specific errors. This was not something I had looked at closely in the first review. The specific recommended changes are in the attached Word document. Please be sure to have the English language reviewer do a final pass before you submit your second revisions.

7. PLOS authors have the option to publish the peer review history of their article (what does this mean?). If published, this will include your full peer review and any attached files.

Reviewer #1: No

---

## [Author Response · Author response to Decision Letter 1]

10 Apr 2021

Response to reviewers 2

Title: Adapting for the COVID-19 pandemic in Ecuador, a characterization of hospital strategies and patients

Authors: Daniel Garzon-Chavez, Daniel Romero-Alvarez, Marco Bonifaz, Juan Gaviria, Daniel Mero, Narcisa Gunsha, Asiris Perez, María Garcia, Hugo Espejo, Franklin Espinosa, Edison Ligña, Mauricio Espinel, Emmanuelle Quentin, Enrique Teran, Francisco Mora, Jorge. Reyes

Reviewer #1

Comment 1. I am very pleased with the authors’ responsiveness to my earlier comments. I appreciate the authors’ taking the time to address all of the comments. I believe the flow of the narrative is much clearer after the reorganization. I do not have major concerns, but I have specific grammar comments, as the PLOS One review system asked reviewers to point out those specific errors. This was not something I had looked at closely in the first review. Please be sure to have the English language reviewer do a final pass before you submit your second revisions. 

Answer: We thank the reviewer for his favorable response to our previous revisions. In this version we have incorporated all the minor changes suggested. 

Comment 2. Line 25: Change “overrun” to “overran.

Answer: Amended (see line 25).

Comment 3. Line 48: Consider changing “rallied” to “amassed”

Answer: Amended (see line 49).

Comment 4. Line 49: Should be “case study” instead of “study case”

Answer: Corrected (see line 50). 

Comment 5. Line 53-54: Consider changing this clause: “Guayaquil allowed mass gatherings and delayed strict isolation around two weeks in relation to Quito” to “Guayaquil banned mass gatherings and implemented strict isolation two weeks later than Quito”

Answer: We thank the reviewer for its detailed correction, which we have applied (see lines 54-55). 

Comment 6. Line 56: Consider changing “Ecuador managed diagnosis of SARS-Cov-2 mainly centralizing real-time reverse transcriptase” to “Ecuador centralized real-time reverse transcriptase”

Answer: Correction applied (see lines 57-58). 

Comment 7. Line 92: Change “considering the likelihood” to “with the likelihood”

Answer: Thanks for this suggestion. We decided to not apply this suggestion because in this case, we are talking about three categories of severity and how they are related with COVID-19 positivity, thus, the word ‘considering’ here is clearly relating both sentences (see line 100-104). 

Comment 8. Section on hospital distribution for COVID-19 patient attendance (114-135): Recommend capitalizing the names of the clinic area and surgery areas in the narrative and also in Figure 1. For example, instead of “clinical area 1” it should be “Clinical Area 1” so readers understand that it is the formal/proper name of those areas. Same goes for “surgery area 1” � “Surgery Area 1”

Answer: We have applied this suggestion in this section, in the figure, and across the main text. Thanks! (see for example lines 132-135 and Fig. 1). 

Comment 9. Line 116: Is “Infection control” a department within the hospital? If so, consider changing to “Infection Control Department”.

Answer: Infection control is a multidisciplinary assembly formed with specialists from different departments. In this version we are calling it Infection Control Unit. We have applied this rewording across the manuscript (see for example line 132 and 169). 

Comment 10. Lines 133, 135, and throughout manuscript: Change all instances of “an score” to “a score”.

Answer: Thank you for noticing this detail. We have corrected it across the manuscript (e.g., see lines 164-165).

Comment 11. Line 144: Consider changing “suspected/positive” to “suspected/confirmed”.

Answer: Amended (see line 175). 

Comment 12. Line 148: Change “got infected” to “was infected”

Answer: We changed the wording to ‘were infected’ considering the meaning of the entire sentence. Thanks! (see line 179).

Comment 13. Line 183-184: Please clarify: Are these people who had laboratory-confirmed SARS-CoV-2 infection or were these simply people who had the RT-PCR done (regardless of results)?

Answer: We are referring to the people that had the RT-PCR test done regardless of the results. Notice that from these 75 patients, 61 are laboratory confirmed positive cases and 14 are laboratory confirmed negative, as explained later in the text. 

Comment 14. Line 190-191: Change “in the context of asymptomatic” to “as asymptomatic”

Answer: Changed as suggested (see lines 234-235). 

Comment 15. Line 193: Here, and throughout, capitalize the proper names of hospital departments, e.g. “Department of Occupational Medicine,” “Infection Control Department”

Answer: Amended (e.g., lines 99, 236-247).

Comment 16. Line 196: Change “to suggest potential clusters” to “to identify potential clusters”

Answer: Fixed (see line 244).

Comment 17. Line 202: Change “and calculated the people at risk” to “and calculated the number of people at risk”

Answer: Fixed (see line 250). 

Comment 18. Line 222: Change “93 positive to SARS-CoV-2” to “93 with laboratory-confirmed SARS-CoV-2 infection”

Answer: We changed ‘93 positive to SARS-CoV-2’ to ‘93 with potential SARS-CoV-2 infection’ since from this 93, 18 patients passed and the remaining 75 are the ones described in the following lines. Thanks for pointing this out. 

Comment 19. Line 225: Change “from both groups” to “overall,” since the overall values are what is being discussed, not the 2 groups separately, as the median was slightly lower for those with a (-) test result.

Answer: Fixed. Thanks (see line 282). 

Comment 20. Line 234: Change “positive patients” to “patients with laboratory-confirmed SARS-CoV-2 infection” or “patients with positive test results” here and throughout. “Positive” is an attribute of the test result or infection status, not of the patients.

Answer: Thanks for this comment. We have amended this mistake across the manuscript (e.g., 273-275, 318). 

Comment 21. Line 236: See comment above. Instead of “positive cases,” use “confirmed cases”.

Answer: We have amended this wording across the entire manuscript. Thanks! (e.g., 273-275, 318).

Comment 22. Line 239: Instead of “negative patients,” use “patients with negative test results”

Answer: Thanks. We have fixed the wording in the entire text (e.g., 273-275, 318).

Comment 23. Line 299: Change “22-health workers” to “22 health workers”

Answer: Fixed (see line 451). 

Comment 24. Perhaps I should have been clearer when I made the recommendation for a more detailed caption. I think the current boldface part of the captions for each table should stay as captions, whereas the non-boldface parts should be table footnotes. What I meant by a more descriptive caption is that captions typically have a descriptor of the sample, the locale (“Quito, Ecuador” or if is the Quito metropolitan area, you may indicate it as “Quito Metropolitan Area, Ecuador”), and some time indicator when the data were collected or sampled (e.g. “February-March 2020”).

For example, for Table 1:

This would be the table caption (note additions highlighted in aqua)

Table 1. General characteristics, risk factors, and symptoms/signs of the first 75 COVID-19

laboratory tested cases attended in the Hospital General del Sur de Quito (HGSQ), Quito, Ecuador -- March 2020

This would be the table footnote

Cases categorized as positive/negative by laboratory testing (i.e., RT-PCR) from 10 to 28 March 2020.

Percentages from the first row are calculated in relation to the total cases (n = 75). Percentages of the

following rows are calculated with positives, negatives, and totals from the first row, respectively.

For Table 2:

This would be the table caption (note additions highlighted in aqua)

Table 2. Blood chemical values, laboratory-confirmed COVID-19 cases (n=61) by intensive care unit (ICU) admission status, Quito, Ecuador -- March 2020

This would be the table footnote

Blood chemical values for 61 COVID-19 positive cases categorized as intensive care unit (ICU; n = 14) and non-ICU (n = 47) from 10 to 28 March 2020. Values represent medians and ranges are depicted in parenthesis. Normal ranges are shown in the headers of each column. CPK = Creatine phosphokinase, LDH = Lactate dehydrogenase, CRP = C-reactive protein, PCT = Procalcitotin, CR = Creatinine; ALT = Alanine transaminase, AST = Aspartate transaminase.

Answer: Thank you very much for your detailed explanation on how to improve the tables. We have applied your suggestions for all the tables (see Tables). 

Comment 25. Table 3. Same comment as above to add a locale and brief sample description to the caption, and separate the non-boldface text to the footnote

Answer: See comment 24. 

Comment 26. Table 4. Same comment as above to add a locale and brief sample description to the caption, and separate the non-boldface text to the footnote

Answer: See comment 24.

Comment 27. Figure 1 A and B. The red arrow on a red box is difficult to see. I recommend changing the color of the boxes to a medium blue (such as this color) instead of red and the color of the arrows to black instead of red. Individuals with color blindness may have an especially hard time seeing the red arrows inside the red boxes.

Answer: We agree that the red arrows are going to be difficult to discriminate. We have changed the color of the arrows and lighted colors of the boxes so everything is more noticeable. We decided to avoid changing the color to the suggested blue because the boxes (rooms) represent infectious sources, and therefore dangerous objects, which traditionally are represented with reddish colors. Thanks! (see Fig. 1). 

Comment 28. Line 324: Change all instances of “sub-employed”/“sub-employment” to “underemployed”/”underemployment”.

Answer: Fixed (see line 479). 

Comment 29. Line 334: Please clarify this line “represents the first attempt to explore the ability of this tool”: the ability of the tool to do what? Also, the first attempt by HGSQ or by whom?

Answer: Thank you very much for this comment; we have reworded the sentence (see lines 488-489).

Comment 30. Line 337: Change “results availability” to “result availability”

Answer: Fixed (see line 492). 

Comment 31. Lines 342-343: Change “poor enough to actually recommend it” to “too poor to recommend”

Answer: Amended (see lines 497-498). 

Comment 32. Line 343: “adverted” is not the correct word here. Perhaps “cautioned” by the Philippine College of Radiology?

Answer: You are totally right. Thank you very much for noticing this miswording. We have changed it as recommended (see line 498). 

Comment 33. Line 345: Awkward wording in “bottlenecks of attention.” Perhaps you meant “bottleneck of patients needing attention”

Answer: We agree with the reviewer, we have changed the sentence (see lines 499-501).

Comment 34. Lines 351-352: Change “since it is impossible to have the luxury of using the only” to “since we cannot justify using the only”

Answer: Amended (see line 520). 

Comment 35. Line 353: Change “exposition” to “exposure”

Answer: Fixed (see line 522). 

Comment 36. Line 385: Change “trials)” to “trials”

Answer: Done (see line 557).

Comment 37. Line 394: Should “ICU attendant” be plural (i.e. “ICU attendants”)?

Answer: Thanks for this comment, we were refereeing to the main director of ICU, we have changed it accordingly (see line 571). 

Comment 38. Line 402: Consider changing “We remark the need” to “Our findings reiterate the need”

Answer: Amended (see lines 578-579). 

Comment 39. Lines 411-413: Consider changing “Hospitals in low and middle-income countries might follow a similar approach regardless of the scientific evidence in favor or against a particular policy because in emergency contexts, evidence-based information lag with respect to actual response needs.” To “Hospitals in low and middle-income countries might follow a similar approach if there is evidence-based information lag with respect to actual response needs.”

Answer: Thanks for this comment. We have made the change as suggested (see lines 589).

Comment 40. Line 415: Change “to remark” to “to note”

Answer: Fixed (see line 591). 

Comment 41. 416: Change “Health sector in Ecuador is fractionated” to “The health sector in Ecuador is fractured”

Answer: Amended. Thanks! (see line 592).

---

## [Decision Letter · Decision Letter 2]

15 Apr 2021

PONE-D-20-15485R2

Adapting for the COVID-19 pandemic in Ecuador, a characterization of hospital strategies and patients

PLOS ONE

Dear Dr. Garzon,

Thank you for submitting your manuscript to PLOS ONE. After careful consideration, we feel that it has merit but does not fully meet PLOS ONE’s publication criteria as it currently stands. Therefore, we invite you to submit a revised version of the manuscript that addresses the points raised during the review process.

Only few grammar errors should be amended by the Authors.

We look forward to receiving your revised manuscript.

Kind regards,

Adriana Calderaro

Academic Editor

PLOS ONE

Journal Requirements:

Reviewers' comments:

Reviewer's Responses to Questions

**Comments to the Author**

1. If the authors have adequately addressed your comments raised in a previous round of review and you feel that this manuscript is now acceptable for publication, you may indicate that here to bypass the “Comments to the Author” section, enter your conflict of interest statement in the “Confidential to Editor” section, and submit your "Accept" recommendation.

Reviewer #1: All comments have been addressed

2. Is the manuscript technically sound, and do the data support the conclusions?

Reviewer #1: Yes

3. Has the statistical analysis been performed appropriately and rigorously? 

Reviewer #1: Yes

4. Have the authors made all data underlying the findings in their manuscript fully available?

Reviewer #1: Yes

5. Is the manuscript presented in an intelligible fashion and written in standard English?

Reviewer #1: Yes

6. Review Comments to the Author

Reviewer #1: Dear authors,

I appreciate the effort you have put in to respond to my suggestions in all rounds of reviews. I have reviewed the most recent draft, and I accept the reasons you provided for not making some of the recommended changes.

I have three more minor grammatical changes to recommend in this round that I missed last time:

Line 393 -- There should not be a comma between "trials" and "assessing"

Line 406 -- Change "referred to" to "reported"

Line 421 -- Change "being" to "be"

The paper should be good to go after this. Again, thank you for being so responsive to the comments. This is an important and timely paper not just for this outbreak for future outbreaks.

Warm regards,

Reviewer 1

7. PLOS authors have the option to publish the peer review history of their article (what does this mean?). If published, this will include your full peer review and any attached files.

Reviewer #1: No

---

## [Author Response · Author response to Decision Letter 2]

15 Apr 2021

Response to reviewers 3

Title: Adapting for the COVID-19 pandemic in Ecuador, a characterization of hospital strategies and patients

Authors: Daniel Garzon-Chavez, Daniel Romero-Alvarez, Marco Bonifaz, Juan Gaviria, Daniel Mero, Narcisa Gunsha, Asiris Perez, María Garcia, Hugo Espejo, Franklin Espinosa, Edison Ligña, Mauricio Espinel, Emmanuelle Quentin, Enrique Teran, Francisco Mora, Jorge. Reyes

Reviewer #1: Dear authors,

I appreciate the effort you have put in to respond to my suggestions in all rounds of reviews. I have reviewed the most recent draft, and I accept the reasons you provided for not making some of the recommended changes.

I have three more minor grammatical changes to recommend in this round that I missed last time:

Line 393 -- There should not be a comma between "trials" and "assessing"

Line 406 -- Change "referred to" to "reported"

Line 421 -- Change "being" to "be"

The paper should be good to go after this. Again, thank you for being so responsive to the comments. This is an important and timely paper not just for this outbreak for future outbreaks.

Warm regards,

Reviewer 1

Answer: We thank the reviewer for his/her efforts in improve the paper and for his/her comments. In this version we have incorporated all the minor changes suggested and we did a detailed examination of the paper. Also, in Line 402 we change “Deparment” to “Department”.

Thank you so much and kind regards.

---

## [Editor Report · Decision Letter 3]

26 Apr 2021

Adapting for the COVID-19 pandemic in Ecuador, a characterization of hospital strategies and patients

PONE-D-20-15485R3

Dear Dr. Garzon,

We’re pleased to inform you that your manuscript has been judged scientifically suitable for publication and will be formally accepted for publication once it meets all outstanding technical requirements.

Kind regards,

Adriana Calderaro

Academic Editor

PLOS ONE
---

## [Editor Report · Acceptance letter]

7 May 2021

PONE-D-20-15485R3 

Adapting for the COVID-19 pandemic in Ecuador, a characterization of hospital strategies and patients 

Dear Dr. Garzon-Chavez:

I'm pleased to inform you that your manuscript has been deemed suitable for publication in PLOS ONE. Congratulations! Your manuscript is now with our production department. 

Kind regards, 

on behalf of

MD, PhD, Associate Professor Adriana Calderaro 

Academic Editor

PLOS ONE